# Computational Pathology at Health System Scale – Self-Supervised Foundation Models from Billions of Images

**Gabriele Campanella[1], Chad Vanderbilt[2], Thomas J. Fuchs[1]**
[1]Icahn School of Medicine at Mount Sinai    [2]Memorial Sloan Kettering Cancer Center

## Abstract

Recent breakthroughs in self-supervised learning have enabled the use of large unlabeled datasets to train visual foundation models that can generalize to a variety of downstream tasks. While this training paradigm is well suited for the medical domain where annotations are scarce, large-scale pretraining in healthcare, and in particular pathology, has not been extensively studied. Previous work in self-supervised learning in pathology has focused on relatively small datasets for both pre-training and performance evaluation of downstream tasks. The aim of this work is to explore foundation models at a scale that goes orders of magnitude beyond the state of the art and benchmark current self-supervised learning algorithms by pre-training and evaluating downstream performance on large clinically relevant pathology tasks.

We compiled the largest academic pathology dataset to date, consisting of over 3 billion images from 423 thousand digital microscopy slides. We compared the pre-training of visual transformer models with focus on masked autoencoders (MAE) and self-distillation models (DINO). Downstream performance is evaluated on six clinically relevant tasks from three anatomic sites and two institutions: breast cancer detection, inflammatory bowel disease detection, breast cancer estrogen receptor prediction, lung adenocarcinoma EGFR mutation prediction, and lung cancer immunotherapy response prediction.

The results demonstrate that pre-training on pathology data is beneficial for downstream performance compared to pre-training on natural images. Additionally, the DINO algorithm achieved better generalization performance across all tasks tested. The presented model performances signify a phase change in computational pathology research, paving the way into a new era of more performant models based on large-scale, parallel pre-training at the billion-image scale.

## Introduction

Artificial Intelligence (AI) is revolutionizing the medical field. The introduction of deep learning[1] has greatly accelerated the development of predictive models for high-dimensional data modalities such as images and text that are not amenable to classical machine learning algorithms. Convolutional neural networks (CNNs) and vision transformers[2] (ViTs) have been used to great effect in a myriad of problems involving supervised learning and have enabled the training of predictive models for a variety of tasks with high performance. Recently, the development of self-supervised learning (SSL) algorithms has marked a paradigm shift by enabling the training of deep neural networks on very large unlabeled datasets, yielding results on par with supervised learning strategies. Large neural networks trained this way can be described as foundation models that can be used for a wide variety of downstream tasks with little to no fine-tuning. Despite the great successes in the computer vision and natural language fields, SSL algorithms and foundation models are still in their infancy in the medical domain. One of the main reasons is the lack of datasets and the necessary computing infrastructure which makes large-scale SSL experiments only possible at large well-funded institutions.

In pathology, the lack of data is even more acute due to the still low adoption of digital pathology. Additionally, digital pathology slides are orders of magnitude larger than other image modalities, with resolutions of tens to hundreds of thousands of pixels in each dimension. This poses challenges in terms of the methods used to analyze the images and the hardware requirements to effectively perform experiments. A common strategy to analyze these images is to divide the slide into tiles and encode them using a deep neural network, expressing the slide as a list of feature vectors and thus reducing the dimensionality of the slide by multiple orders of magnitude. In a second step, the feature vectors are aggregated using a neural network to obtain a slide-level representation. The first step is by far the most computationally expensive, while the second step requires much fewer resources. This is why most studies in computational pathology rely on already existing pretrained encoders, usually trained on natural images and not pathology. There is a need for strategies that enable training of encoders directly on pathology images, and SSL lends itself well for this task as it does not require any sort of labels and could allow for the training of a pathology foundation model on large datasets. SSL for pathology has recently received lots of attention, and there are many academic and non-academic efforts to generate a general-purpose pathology model[3–9]. We present a summary of notable works in this area in the Supplementary Related Works section.

It is becoming abundantly clear that using SSL to train image encoders on unlabeled pathology data is superior to

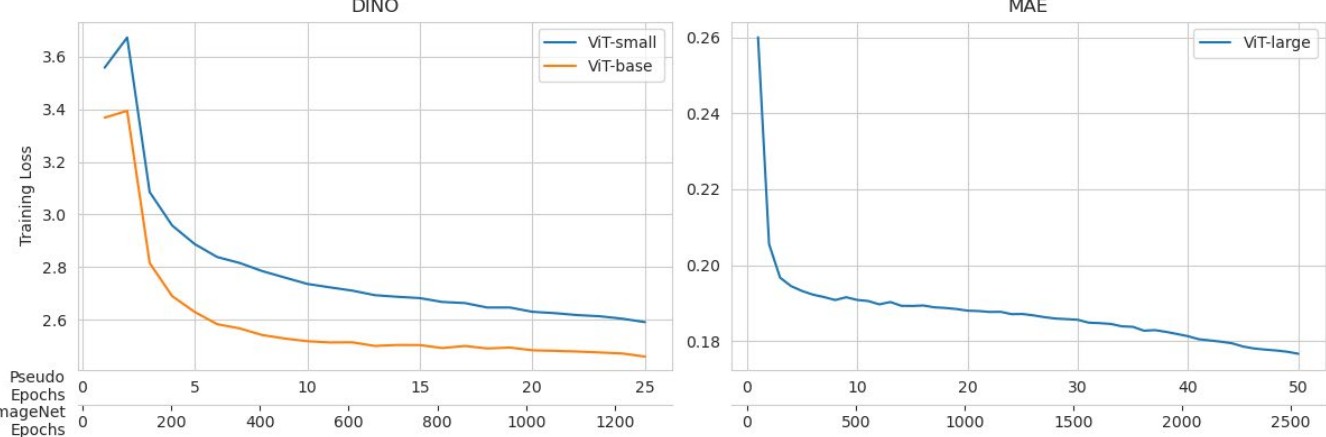

Figure 1. Training curves for the SSL algorithms: left - DINO with a ViT-small and ViT-base models, right - MAE with a ViT-large model. On the y axis the respective training loss is reported. On the x axis pseudo-epochs and ImageNet epochs are shown.

relying on models pretrained on other domains such as natural images. While general vision SSL methods and pathology specific SSL methods hold immense potential, there are still some challenges to be overcome before pathology foundation models can be used reliably in clinical workflows. Datasets used to train pathology models are still very small compared to other domains. While there are many studies in radiology that include millions of images[10,11], the majority of SSL studies for pathology to date are based on TCGA which consists of around thirty thousand slides only. Given the evidence from the natural language and vision domains that larger datasets and higher capacity models will produce better performance especially in the SSL setting, training on larger pathology datasets should be a priority. Furthermore, the downstream performance of SSL models for pathology is rarely assessed on clinically oriented tasks. Tile-based predictions, organ classification, coarse segmentation, captioning, retrieval, and VQA are valuable scientific explorations, but less relevant in the clinical setting. This effect is compounded by the use of curated public datasets which may not be suited for assessing generalization to real world data. Downstream performance should be evaluated on clinical data, preferably from multiple institutions, for clinically relevant tasks such as diagnostic assessment, biomarker prediction, and outcome prediction.

To overcome these limitations, we introduce a library of models trained on the largest pathology dataset to date consisting of over 3 billion images from over 420 thousand clinical slides from a large health system. In this work we strive to benchmark state-of-the-art SSL algorithms on a variety of clinically relevant tasks, anatomic sites, and disease indications. Our aims are to determine the best training strategies for foundation models in pathology, and to share with the research community resources for embedding and benchmarking. To this point we have trained various ViT variants

with DINO[12] and MAE[13] and analyzed their performance on six clinically relevant tasks from three anatomic sites.

## Foundation Model Pre-Training

To perform pre-training of pathology foundation models we leveraged a large clinical dataset consisting of 423,563 slides from 88,035 cases and 76,794 patients. The dataset included slides from 70 distinct organs across all the specialties of pathology. The total storage required for the compressed image files was around 600TB. The presented dataset is one order of magnitude larger than any previous effort in computational pathology. For training we constructed pseudo-epoch consisting of 65 million tiles from one fifth of the slides, equivalent to 50.7 ImageNet epochs[14]. Hence, five pseudo-epochs are needed for a complete pass through every slide in the dataset. The SSL algorithms were cloned directly from their official GitHub repositories. No changes were made to the code except for customizing the data loading procedures. Models were trained in parallel on at least 8 GPUs, and up to 24 GPUs, depending on cluster availability. Further details can be found in the Supplementary Methods section. The SSL algorithms tested so far include i) DINO[12], a self-distillation-based algorithm and ii) MAE[13], a masked image modeling-based algorithm. The following models were trained: i) A ViT-small (21.7M parameters, 384 feature dimensionality) with DINO for 25 pseudo-epochs on 12

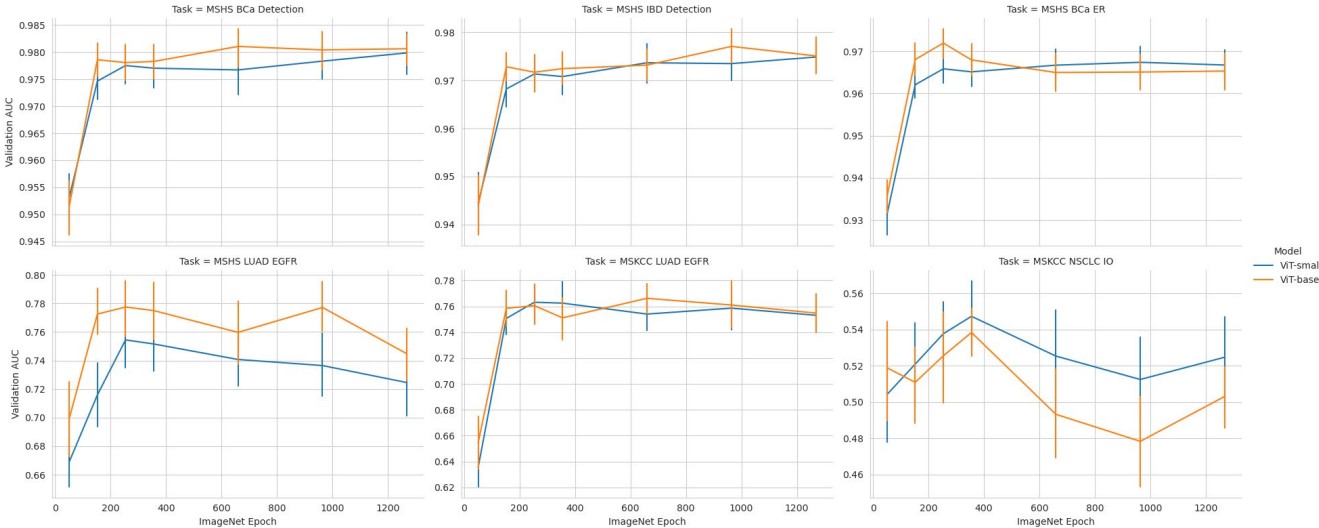

Figure 2. Downstream performance for the DINO models at various checkpoints during training. Each plot shows the distribution of the MCCV results. Tasks from top left to bottom right are: i) MSHS breast cancer detection, ii) MSHS IBD detection, iii) MSHS breast cancer ER prediction, iv) MSHS lung cancer EGFR prediction, v) MSKCC lung cancer EGFR prediction, vi) MSKCC lung cancer immunotherapy outcome prediction. It is interesting to note that downstream performance starts to saturate early during training, after around 5 pseudo-epochs.

A100 40GB GPUs with batch size of 90 per GPU. Training was completed in roughly 17 days and 16 hours. ii) A ViT-base (85.8M parameters, 768 feature dimensionality) with DINO for 25 pseudo-epochs on 8 H100 80GB GPUs with batch size of 100 per GPU. iii) A ViT-large (303.3M parameters, 1024 feature dimensionality) was trained with MAE for 50 pseudo-epochs on 8 A100 40GB GPUs with batch size of 180 per GPU. Training was completed in roughly 17 days and 10 hours.

Figure 1 shows the training curves of the pre-training experiments in terms of pseudo-epochs and ImageNet equivalent epochs.

## Clinical Benchmarking Results

To assess generalization performance to downstream tasks, we collected six datasets from two institutions: Mount Sinai Health System (MSHS) and Memorial Sloan Kettering Cancer Center (MSKCC). Across the two institutions slides were scanned on Philips Ultrafast scanners and Leica Aperio AT2 scanners. The tasks included are:
• MSHS breast cancer (BCa) detection cohort, 1,998 slides (999 positive and 999 negative).
• MSHS BCa ER prediction cohort, 2,000 slides (1,000 positive and 1,000 negative).
• MSHS inflammatory bowel disease (IBD) detection cohort, 1,441 slides (717 with active inflammation and 724 with normal mucosa).
• MSHS EGFR mutation detection in Lung Adenocarcinoma (LUAD), 294 slides (103 positive and 191 negative).

• MSKCC EGFR mutation detection in LUAD, 1,000 slides (307 positive and 693 negative).
• MSKCC non-small cell lung cancer (NSCLC) immunotherapy response prediction, 454 slides (86 positive and 368 negative).
More detailed information on the tasks can be found in the Supplementary Methods section.

Downstream task performance was assessed by training a Gated MIL Attention (GMA)[15] slide aggregation model with a linear classifier on top. Since GMA does not consider the spatial distribution of tiles over the slide in its prediction, it is a simple method to test the expressiveness of the feature space generated by the SSL pretraining akin to linear probing used in traditional SSL literature. Generalization performance was estimated using 20 rounds of Monte Carlo Cross-Validation (MCCV) for each model and task. As a baseline we used an ImageNet pre-trained truncated ResNet50 as in Lu et. al.[16].

Convergence curves summarizing the comparison between baselines and trained SSL models are presented in Supplementary Figure 1. Not surprisingly, detection tasks achieved high performance with AUCs over 90%. Biomarker prediction tasks exhibited more variability in the performance depending on the biomarker in question. The outcome prediction task included yielded poor results across all tested models. The models trained using DINO showed superior performance across all downstream tasks, except for outcome prediction, where all strategies resulted in a performance close to chance. It is interesting to note that the ResNet50 experiments showed signs of overfitting in every

task, possibly due to the higher dimensionality of the embeddings. If considering using a ResNet50, it may be beneficial to use early stopping. Conversely, the features extracted from the ViT-small and ViT-base models trained with DINO led to faster convergence without signs of overfitting.

To investigate the importance of dataset size, we analyzed the downstream performance of the features extracted from the DINO models at various checkpoints during pre-training. The results of this analysis are presented in Figure 2. It can be observed that saturation of downstream performance tends to occur early during training. In most cases, after 5 pseudo-epochs performance reaches its maximum or it is close to it. Longer pre-training led to small or no improvements, and in some cases, the performance degraded on downstream tasks. It is interesting to note that 5 pseudo-epochs (also equivalent to 325 million tiles) correspond to a full pass through all the slides in the pre-training dataset. It may be that once all the slides have been included in the training, additional training passes result in diminishing returns.

## Discussion

Self-Supervised Learning and in general large-scale pre-training have obtained results that were unimaginable only a few years ago. We expect that the scaling rules observed for SLL experiments in computer vision and natural language processing will apply in the context of pathology as well. In this study we explored the use of two SSL algorithms to pre-train vision models on the largest pathology dataset to date. Compared to previous efforts to train pathology foundation models, our work presents a pre-training dataset that is at least one order of magnitude larger and consists entirely of slides produced during routine clinical workflows without any data curation. Additionally, the downstream performance of these models was tested on clinically relevant tasks spanning diagnosis, biomarker, and outcome prediction. Our results support the idea that SSL-based pre-training is able to extract relevant morphological information from histology and achieves better performance than other pre-training strategies that do not involve pathology images.

Despite the evidence presented here, many important questions still have to be addressed concerning many aspects of such experimental set-ups: pre-training data, encoder architecture, training algorithm, and downstream tasks. In terms of the dataset, we observed that downstream performance plateaued relatively early during pre-training. Investigating how the dataset size and variability influences the generalization performance could lead to more efficient data strategies. How many slides and how many tiles are required for optimal performance are important questions that

we will address in future iterations of this work. Tightly linked to the dataset size is the choice of encoder architecture. Due to lack of resources, we only trained a small ViT variant with the DINO algorithm. Smaller architectures may saturate at lower data regimes. As we increase our dataset sizes, larger models will become necessary. While our results indicate that the off-the-shelf DINO strategy can boost performance over natural image pre-training, other strategies may be better suited to pathology data and some algorithms explicitly take advantage of the characteristics of pathology images. We plan to expand the number of algorithms benchmarked in this project, which will allow us to determine optimal training strategies for SSL in pathology. Lastly, while we strived to include as many clinically relevant tasks as possible, there is much room for improvement. We will include more tasks and critically, more organs and diseases, with more emphasis on more challenging tasks such as outcome prediction. Furthermore, the inclusion of diverse benchmarking cohorts will enable us to study whether biases can arise when developing predictive models based on pathology data.

As a final note, the computational pathology field would benefit from the release of publicly available pathology data that goes beyond small and curated datasets. A public benchmark of clinically relevant tasks would allow for algorithm and model comparisons with better generalization potential. As AI and clinical data have become highly valued, it is unlikely that the current lack of public datasets will change. In a parallel with large language models, where only large companies have the resources to train them at scale, only well-funded health systems which have digitized their pathology departments and have amassed large amounts of digital slides will be able to train clinically sound pathology foundation models. In the interest of the community, we are working towards building a service and an API for users to: i) produce embeddings from our trained models and ii) benchmark their models on our clinical tasks. It is our hope that this effort will boost computational pathology research and facilitate the development of clinical-grade decision support systems.

## Acknowledgements

This work is supported in part through the use of research platform AI-Ready Mount Sinai (AIR.MS), the expertise provided by the team at the Hasso Plattner Institute for Digital Health at Mount Sinai (HPI.MS), and through the computational and data resources and staff expertise provided by Scientific Computing and Data at the Icahn School of Medicine at Mount Sinai and supported by the Clinical and Translational Science Awards (CTSA) grant UL1TR004419 from the National Center for Advancing Translational Sciences.

## Ethics Board Approval

This study has been approved by the Mount Sinai Institutional Review Board (IRB #19-0051).

## Supplementary Related Works

Wang et al.[3] proposed SRCL, an SSL method based on MoCo v3[17], along with CTransPath, a model architecture that combines convolutional layers with the Swin Transformer[18] model. They trained their model on 15.6 million tiles from 32,220 slides from the TCGA and PAIP datasets spanning 25 anatomic sites and over 32 cancer subtypes. The downstream performance was assessed on patch retrieval, supervised patch classification, weakly-supervised WSI classification, mitosis detection, and colorectal adenocarcinoma gland segmentation. Methodological advances include the introduction of a strategy to sample positive examples for the contrastive approach, and the hybrid convolutional-transformer model architecture.

Kang et al.[4] analyzed the performance of four SSL methods (MoCo v2[19], SwAV[20], Barlow Twins[21], and DINO[12]) applied to pathology data. They sourced 19 million patches from 20,994 TCGA slides to train their models. In contrast to other works which focus on 20x magnification, they used both 20x and 40x. Downstream task performance was evaluated on four tile-level image classification tasks and one nuclei segmentation task. The main contribution in terms of methodology is the optimization of augmentation strategies for histology data.

Filiot et al.[5] analyzed the performance of iBOT[22], an SSL framework that combines masked image modeling and contrastive learning, on histology data. They trained several ViT models on a dataset consisting of up to 43.3 million tiles from 6,093 TCGA slides and 13 anatomic sites. They assessed the performance of learned features on 17 downstream tasks across seven cancer indications including tile-level and slide-level tasks for subtype, genomic alteration, and overall survival prediction.

Lu et al.[6] proposed CONCH, a pathology foundation model framework that features a vision-language joint embedding space. They first trained a ViT on a sample of 16 million tiles from 21,442 proprietary in-house WSIs using the iBOT[22] SSL framework. Further, based on the previous ViT backbone, they trained a visual-language model based on CoCa[23] using 1.17 million image-caption pairs originating from educational resources and PubMed articles. The capabilities of the model were tested on 13 downstream tasks including tile and slide classification, cross-modal image-to-text and text-to-image retrieval, coarse WSI segmentation, and image captioning.

Tu et al.[7] proposed a proof of concept for a generalist biomedical AI system. They first introduced MultiMedBench, a collection of 12 open-source datasets for a variety of tasks that span language and various imaging modalities. One of the tasks included is Path-VQA[24], a visual question answering (VQA) dataset for pathology consisting of 4,998 tiles associated with over 30 thousand question-answer pairs. They then finetuned a PaLM-E[25] model on the Multi-MedBench set. The result is Med-PaLM-M, a multi-modal system that can analyze inputs coming from various medical data modalities.

Chen et al.[8] introduced UNI, a ViT-large model trained on 100 thousand proprietary slides using the DINOv2[26] SSL algorithm. The pre-training dataset they used included 100 million tiles from 20 major tissue types. They assessed the downstream performance on 33 tasks including tile-level tasks for classification, segmentation, and retrieval and slide-level classification tasks.

Vorontsov et al.[9] introduced Virchow, a ViT-huge model trained on 381 million tiles coming from almost 1.5 million proprietary slides with DINOv2. Slides from 24 tissue types were included. Downstream task performance was assessed on tile-level and slide-level benchmarks including tissue classification and biomarker prediction.

## Supplementary Methods

### Pre-Training Datasets

For pretraining purposes, we collected the largest digital pathology dataset to date, consisting of 423,563 H&E-stained slides from 88,035 cases and 76,794 patients. The dataset included slides from 70 distinct organs across all the specialties of pathology. All slides were scanned on a Philips Ultrafast scanner at 0.25 microns per pixel (MPP) resolution, de-identified and converted to tiff format. The total storage required for the raw tiff files was around 600TB. As a preprocessing step, tissue tiles were extracted from each slide at 0.5 MPP resolution. The total number of tiles generated was over 3 billion, for a total storage requirement of about 100TB. To put this into perspective, the largest study to date on SSL for pathology used over 1.5 million slides, but sampled less than 400 million tiles[9]. The largest dataset described in an academic setting included 100 million tiles from 100 thousand slides[8]. The presented dataset is one order of magnitude larger than any previous effort in computational pathology.

To allow for reproducible experiments, the tile-level training schedule was hardcoded. Due to unbalanced organ frequencies in the dataset, tiles were sampled from slides based on their organ to obtain a more balanced representation of all organs. For each pseudo-epoch, 65 million tiles were sampled from one fifth of the slides, where each pseudo-epoch corresponds to 50.7 ImageNet epochs[14]. Hence, five pseudo-epochs are needed for a complete pass through every slide in the dataset. Training schedules were

hardcoded for 50 pseudo-epochs for a total of 10 passes through the 423 thousand slides and 3.2 billion tiles. It is important to note that training durations will be different for different experiments due to constraints in computing resources.

## Computing Infrastructure and Software

MSHS's HPC cluster utilizes 24,214 Intel Platinum compute cores. It includes 22 GPU nodes: 12 V100 GPU nodes and 10 A100 GPU nodes. It includes 32 petabytes of spinning storage accessed via IBM's Spectrum Scale/General Parallel File System (GPFS) for a total of 2 petaflops of compute power. Experiments were conducted on GPU nodes using python and the pytorch[27] library. Pytorch's distributed data parallel (DDP)[28] was used to run multi-node multi-gpu workloads.

## Baselines

Performance on downstream tasks of the SSL trained models were compared to popular slide analysis strategies from the computational pathology field. We extracted features from a truncated ResNet50 (8.5M parameters, 1024 feature dimensionality) pretrained on ImageNet as in Lu et al.[16]. Additionally, we also extracted features from a ResNet50 (23.5M parameters, 2048 feature dimensionality) pretrained on ImageNet.

## Self-Supervised Pre-Training

The SSL algorithms were cloned directly from their official GitHub repositories. No changes were made to the code except for customizing the data loading procedures. Models were trained in parallel on at least 8 GPUs, and up to 24 GPUs, depending on cluster availability. The SSL algorithms tested so far include i) DINO[12], a self-distillation-based algorithm and ii) MAE[13], a masked image modeling-based algorithm. The following models were trained:

- A ViT-small (21.7M parameters, 384 feature dimensionality) with DINO for 25 pseudo-epochs on 12 A100 40GB GPUs with batch size of 90 per GPU. Training was completed in roughly 17 days and 16 hours.
- A ViT-base (85.8M parameters, 768 feature dimensionality) with DINO for 18 pseudo-epochs on 8 A100 80GB GPUs with batch size of 100 per GPU.
- A ViT-large (303.3M parameters, 1024 feature dimensionality) was trained with MAE for 50 pseudo-epochs on 8 A100 40GB GPUs with batch size of 180 per GPU. Training was completed in roughly 17 days and 10 hours.

## Benchmark Datasets

To assess generalization performance to downstream tasks, we collected six datasets from two institutions. The Mount Sinai Health System's slides were scanned on Philips Ultrafast scanners, while the slides from Memorial Sloan Kettering were scanned on Leica Aperio AT2 scanners.

MSHS breast cancer (BCa) detection cohort. Breast cancer blocks and normal breast blocks were obtained from the pathology LIS. A total of 1,998 slides were sampled, 999 positive and 999 negative. The positive slides were selected from blocks that received the routine biomarker panel for cancer cases (estrogen receptor ER, progesterone receptor PR, HER2, and Ki67), while negative slides were selected from breast cases that did not have an order for the routine panel. Additionally, negative cases were selected if they were not a mastectomy case, did not have a synoptic report associated with the case, and had no mention of cancer or carcinoma in the report.

MSHS 1 BCa Estrogen Receptor (ER) prediction cohort. Breast cancer cases with orders for ER IHC were queried from the LIS. The IHC result was automatically extracted from the pathology report. A total of 2000 slides were sampled, 1000 positive, 1000 negative.

MSHS Inflammatory Bowel Disease (IBD) detection cohort. Normal mucosa samples were obtained from patients

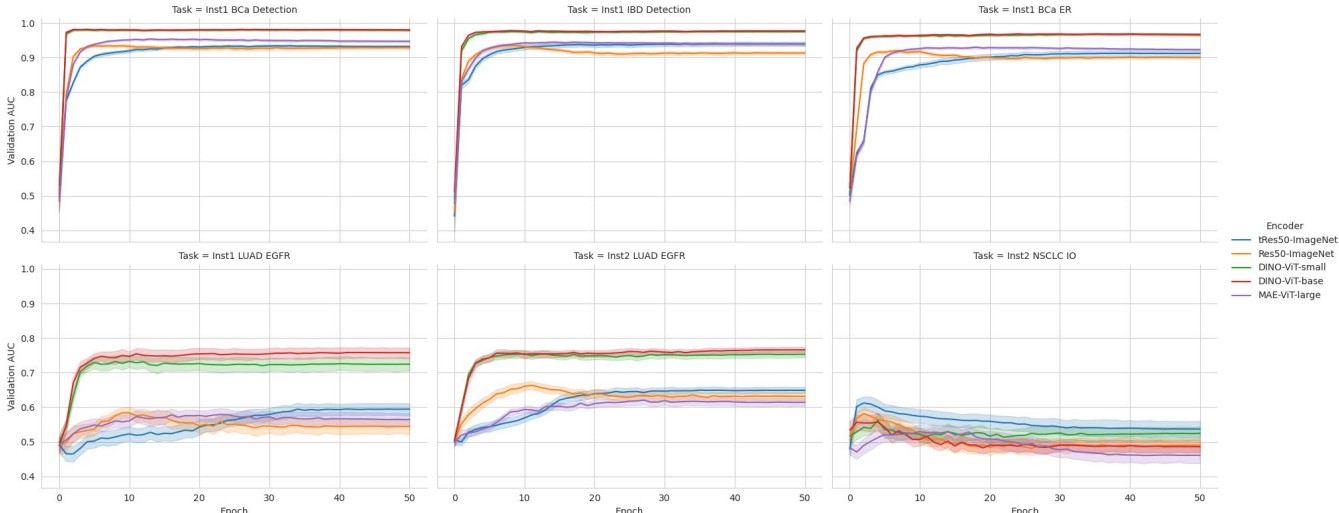

Supplementary Figure 1. Benchmark task results. Each panel depicts the training results for a specific task, comparing baselines and SSL models. Each line summarizes the training of a GMA aggregation model on 20 MCCV splits with 2 replicas each. The solid line represents the average, while the shaded area represents the 95% confidence interval (CI) computed via bootstrapping with 1000 iterations. Tasks from top left to bottom right are: i) Institution 1 breast cancer detection (1,998 slides), ii) Institution 1 IBD detection (1,441 slides), iii) Institution 1 breast cancer ER prediction (2,000 slides), iv) Institution 1 lung cancer EGFR prediction (294 slides), v) Institution 2 lung cancer EGFR prediction (1,000 slides), vi) Institution 2 lung cancer immunotherapy outcome prediction (454 slides). In general, we observe a superior performance from the DINO trained model with significant improvement for biomarker prediction. The outcome prediction task is the exception where all models result in poor performance.

undergoing screening and routine surveillance lower endoscopy from 2018 to 2022. IBD cases, including first diagnoses and follow ups, were included. Active IBD samples were scored using the Mount Sinai histologic disease criteria and found to have Histologic Activity Score (HAI) $\geq 1$[29]. A total of 1441 slides were sampled, 717 with active inflammation and 724 with normal mucosa.

MSHS EGFR mutation detection in Lung Adenocarcinoma (LUAD). A total of 294 slides were obtained from the clinical slide database, 103 positive and 191 negative. The cohort was built following the guidelines described in Campanella et al.[30] to map mutations to a binary target.

MSKCC EGFR mutation detection in LUAD. This is a sample of the dataset described in Campanella et al.[30]. A total of 1000 slides were sampled at random, 307 positive and 693 negative.

MSKCC lung cancer immunotherapy response prediction. Non-small cell lung cancer (NSCLC) patients who received PD-L1 blockade-based immunotherapy between 2013 and 2019 at MSKCC were considered. Cytology specimens were excluded. Objective overall response was determined by RECIST[31] and performed by a blinded thoracic radiologist. A total of 454 slides were obtained, 86 positive and 368 negative.

## Benchmark Training

In the SSL literature, the performance of downstream tasks is frequently assessed by training a linear classifier (linear probing) on top of features extracted by the frozen encoder, or via zero-shot approaches such as k-NN. For pathology slides, there is no direct way to translate these approaches without having tile-level annotations. To overcome this challenge, we trained a slide-level aggregator based on the popular Gated MIL Attention (GMA) model[15] with a linear classifier on top. Since GMA does not consider the spatial distribution of tiles over the slide in its prediction, it is a simple method to test the expressiveness of the feature space generated by the SSL pretraining.

To estimate generalization performance, we employed a Monte Carlo Cross-Validation (MCCV) strategy where for each MCCV split, 80% of the samples were assigned to the training and the rest to validation. For each benchmark task the 20 MCCV folds were randomly sampled and kept fixed for all experiments. Each MCCV split was run twice to assess stochastic fluctuations during training. All models were trained with a single V100 GPU for 50 epochs using the AdamW[32] optimizer. A cosine decay with warm up schedule was used for the learning rate and weight decay hyperparameters. For reporting the experimental results, line plots of the convergence of loss and AUC show all 40 runs (20 MCCV splits, 2 replicas) for each experiment. Line plots

summarizing the results for each experiment are constructed by taking the final validation AUC after training and averaging the replicas of the same MCCV split.

## Supplementary Results

### Clinical Benchmarking Results

Convergence curves summarizing the comparison between baselines and trained SSL models are presented in Supplementary Figure 1.

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
