# OpenReview forum: "Computational Pathology at Health System Scale – Self-Supervised Foundation Models from Billions of Images"
_AAAI.org/2024/Spring_Symposium_Series/Clinical_FMs — AAAI 2024 SSS on Clinical FMs_

### Official Review · Reviewer_Dzb1 · 2024-02-15
**A good start to potentially impactful work**

**Rating:** 7
**Confidence:** 4

**Review:**

The authors train self-supervised vision foundation models on a very large set of pathology data and show that the embeddings they produce yield much better performance on downstream tasks compared to those trained on general image data, specifically ImageNet. The writing is clear, the presentation is thorough, and the results are strong. The work has potential for broad impact. I didn't read the supplementary sections in detail, but the thoroughness is appreciated. I am not familiar with pathology, so I will leave it to other reviewers to assess the selection of downstream tasks.

There are elements of the experiments and presentation that could be improved. While it should be straightforward to improve presentation, I understand that, since the experiments themselves are costly to run, additional experimental runs may not be able to be included in this submission, which is fine - it can be taken as feedback for further development of the work.
1. It is not necessary to show all the results across epochs. It makes the figures unnecessarily large and difficult to interpret, and it masks the effect of model selection, e.g. using a validation set to determine which checkpoint's model to actually use. In particular, the overlapping lines make Supplementary Figure 1 a bit hard to read. Perhaps just one figure to make the point about saturation and overfitting would be enough. The rest could be tables or bar charts or similar for the selected models.
2. It does not seem appropriate to draw conclusions about the effect of data quantity from the loss curves shown. I don't think you can reasonably disentangle the effects of training time and data quantity in these results. The best approach would be to train additional models on subsamples of the data set, but I understand this is costly.
3. It's unclear why not all models are shown in Figure 2.
4. Supplementary Figure 1 contains the main results of the study. These should be in the main paper. Just a table would be fine.
5. It's not clear what is tRes50 vs Res50.
6. The baseline model should have the same architecture (ViT) as the experimental models in order to isolate the effect of the pre-training data. It is even acknowledged by the authors that ResNet may be overfitting due to the architecture itself.
7. It is explained that DINO-ViT-large is excluded due to training cost, but why is there no MAE-ViT-small or MAE-ViT-base?
8. I would expect that the data cannot be released, but why can the pretrained models not be released? Regardless, the intention to set up an API to get embeddings is appreciated.

Minor nitpicks:
1. In the pre-training section, you mention that your data is an order of magnitude larger than any previous effort. It would be nice to cite the largest previous effort here.
2. Please define pseudo-epoch and explain why it is used instead of standard epochs (I am guessing it is to increase checkpoint frequency?).
3. Typo, first paragraph of discussion section: "SLL"
4. In the discussion section, you say you trained DINO only on ViT-small, but you report results also for ViT-base.

---

### Official Review · Reviewer_G7XQ · 2024-02-17
**The paper presents a study on self-supervised learning for computational pathology, utilizing large-scale datasets and ViT models, demonstrating superior performance on clinically relevant tasks.**

**Rating:** 8
**Confidence:** 4

**Review:**

The paper presents a comprehensive study on the application of self-supervised learning (SSL) in computational pathology, focusing on the pre-training and downstream performance evaluation of visual foundation models on large-scale pathology tasks. The study compiled a massive academic pathology dataset, consisting of over 3 billion images from 423 thousand digital microscopy slides, and compared the pre-training of visual transformer models using masked autoencoders (MAE) and self-distillation models (DINO). The downstream performance was evaluated on six clinically relevant tasks from three anatomic sites and two institutions, demonstrating the benefits of pre-training on pathology data for downstream performance compared to pre-training on natural images. The DINO algorithm achieved better generalization performance across all tasks tested, signifying a significant advancement in computational pathology research.

Pros:

The study addresses a critical gap in the application of SSL algorithms and foundation models in the medical domain, particularly in computational pathology.

The compilation of the largest academic pathology dataset to date, consisting of over 3 billion images, demonstrates a significant contribution to the field.

The comparison of pre-training methods and evaluation of downstream performance on clinically relevant tasks provides valuable insights for the development of performant models in computational pathology.

The study's findings indicate a phase change in computational pathology research, paving the way for more performant models based on large-scale, parallel pre-training at the billion-image scale.

Cons:

The study could benefit from a more detailed discussion on the limitations and challenges of SSL algorithms and foundation models in the medical domain, particularly in clinical workflows.

While the downstream performance was evaluated on clinically relevant tasks, the study could further emphasize the potential impact of these findings on real-world clinical applications.

The document lacks a detailed discussion on the ethical considerations and potential biases associated with the use of large-scale pathology datasets and SSL algorithms in healthcare.

Overall, the work demonstrates high quality, clarity, originality, and significance in advancing the application of SSL algorithms and foundation models in computational pathology. The study's comprehensive approach, large-scale dataset compilation, and valuable insights into pre-training methods and downstream performance evaluation contribute significantly to the field of computational pathology. However, further discussion on ethical considerations and potential biases, as well as the translation of findings into real-world clinical applications, would enhance the overall impact of the work.

---

### Official Review · Reviewer_cUiu · 2024-02-22
**A Large Scale Self Supervised FM for Computational Pathology**

**Rating:** 6
**Confidence:** 3

**Review:**

**Summary**:

The authors present comprehensive work towards building a Foundation Model for computational pathology. They motivate the need for an FM in pathology, providing a background of existing work in the field. They present three models based on the Visual Transformer Architecture combined with two SSL algorithms, DINO and MAE.  They have compiled a significant dataset for the pretraining of their FM using SSL and perform benchmarking on multiple downstream tasks. Their approach shows higher AUC for most tasks over the baselines they have used.

**Pros**:

- The models proposed by the authors showcase a clear superiority in performance to the baselines.
- The authors have collected an impressive amount of data pre-training their FM, alluding to their collected dataset being an order of magnitude larger than any other data collected in the field.
- The authors have done a good job of investigating the training behavior of their models, and have indicated potential next steps for extending their work, all of which I agree with.
- I am glad the authors have discussed open-sourcing their model, as I view their data collection and FM as valuable contributions to the field of pathology.

**Considerations**:

- Can the authors provide some reasoning regarding the generally poor performance observed across the proposed models and baselines for Task 6: `Institution 2
lung cancer immunotherapy outcome prediction`? It appears that this dataset has the largest label imbalance across the different tasks the authors are testing for. Could that be the reason?
- Authors use validation AUC to indicate performance, but AUC as a metric captures an aggregate performance of the model across different operational thresholds. When operationalizing an FM for a clinical setting, one often faces the dilemma of considering the ideal operational threshold of classification (especially in the binary case which is the case with a lot of the downstream tasks the authors test their models on). This is a minor nitpick, and maybe something the authors can show in supplementary material. However, I would be interested in the tradeoffs their FM makes on sensitivity vs specificity at a given threshold for the various downstream tasks.
- It is interesting to note that ViT-large with MAE performs worse in most cases than the two ViT models trained with DINO (in three cases, worse than the baseline). Can the authors comment on why they think this is? Did they explore ViT-large with DINO? If not, could they comment on why?
- The authors have provided multiple examples of SSL models pre-trained on pathology data, could they comment on why they didn't use some of those methods as baselines along with ResNet50?

**Quality**:

The overall quality of the paper is good.

**Originality**:

The authors pre-train their models on a very large corpus of pathology data, which they indicate is larger than any corpus of pathology data collected before. While the authors have used some off-the-shelf methods like DINO for their SSL strategy, the scale of the data they have pre-trained their models on encourages me to believe in the novelty of their SSL approach. This is further reflected by excellent performance in downstream tasks with their proposed approach. However, I am a little concerned with the lack of variety in their chosen baselines, I would like the authors to add some more baselines that use SSL as a pre-training strategy to firmly indicate the superiority of their SSL approach.

**Significance**:

The authors' contributions to the field of pathology with their FM and their collected corpus of data could potentially be very significant for the field of pathology. The authors have correctly identified a list of follow-up questions based on their approach which could further help assert the significance of their FM if they are answered.

**Miscellaneous Comments**:

- Could the authors elaborate a little more about GMA, as this is not a method I am familiar with? My assumption was the spatial distribution of the tiles of a single slide would be necessary for the downstream prediction of the slide as a whole, since you do not have tile-level annotations.  Yet. the authors state in benchmark training that GMA does not consider the spatial distribution of the tiles in its prediction. I would appreciate it if the author clarified why GMA's property of not considering the spatial distribution of tiles works here.

---

### Official Review · Reviewer_7k2t · 2024-02-24
**Interesting work but corrections are required**

**Rating:** 7
**Confidence:** 5

**Review:**

The work "Computational Pathology at Health System Scale – Self-Supervised Foundation Models from Billions of Images" is an interesting report from the conducted experiments. The reviewer would like to claim that the problem stated by the Authors is of high interest to broader public (benchmark of recent Foundation Models on pathology dataset). The important thing is that the Authors trained their models with an extensive dataset (as it was claimed in the paper there was around of 3 billion images from around 423000 digital microscopy slides). The goal of the work is clearly given. However, the reviewer would like to raise two suggestions that need to be addressed before publication:
1. The first one is related to the description of the samples. It was claimed that all of them belong to 76794 patients - but no sufficient details about the patients are given. I mean information about the sex, race, age... etc. All these details can allow reader to better understand the approach (of course, I am totally aware that they could not have any alignment with the dataset itself but may have - as some of the illnesses are more probable in later stages of life).
2. I do not understand why huge amount of information is given in the form of supplementary material. I assume that all these subchapters need to be provided directly into the paper - not in the form of supplementaty material. It will be then easier to understand the whole idea as well as to compare the outcomes with the latest results.

The reviewer would like to claim that after all these corrections, the work is ready for publication.